# Analysis of Health-Related Behaviors of Adult Korean Women at Normal BMI with Different Body Image Perceptions: Results from the 2013–2017 Korea National Health and Nutrition Examination Survey (KNHNES)

**DOI:** 10.3390/ijerph17155534

**Published:** 2020-07-31

**Authors:** Seyeon Park, Jieun Shin, Seunghui Baek

**Affiliations:** 1Department of Nursing, Daejeon Health Institute of Technology, 21 Chungjeong-ro, Dong-gu, Daejeon 34504, Korea; sy_park@hit.ac.kr; 2Department of Liberal Arts, Woosuk University, 443 Samnye-ro, Samnye-eup, Wanju-gun, Jeollabuk-do 55338, Korea; 3Department of Health Exercise Management, Sungshin Women’s University, Bomun-ro 34da-gil, Seongbuk-gu, Seoul 02844, Korea

**Keywords:** body image, health behavior, body mass index, Korean women, Korea National Health and Nutrition Examination Survey (KNHNES)

## Abstract

The tendency of misperceiving one’s body image was found to be higher among those at normal body mass index (BMI). Thus, the present study aims to provide basic data to seek solutions for ideal physical activities and right body image perception by comparing health-related behaviors of women at normal BMI. Among the 39,225 respondents from the Korea National Health and Nutrition Examination Survey (KNHNES) conducted from 2013 to 2017, 10,798 adult women with World Health Organization (WHO) BMI Classifications of 18.5 ≤ BMI < 25 (Asia-Pacific) were considered, from which pregnant and breast-feeding women and women whose body image perception was not identified were excluded, leading to a total of 9288 women. Data were analyzed utilizing SAS ver. 9.4 for frequency analysis, cross tabulation, GLM (generalized linear model), and logistic regression analysis with complex samples design, in conformity with the guidelines of the KNHNES. The results showed that approximately most (87.6%) of adult Korean women misperceived their body image. Misperception of body image was related to inappropriate health-related behaviors such as smoking, insufficient sleeping, and excessive body weight management; those who had underestimated their body image (≤64 odds ratio (OR) (0.718 (confidence interval (C.I.) 0.594–0.866))) carried out fewer health-related behaviors, while women aged 65 or above engaged in more health-related behaviors when they perceived themselves as obese (OR 1.683 (C.I. 1.260–2.248; overestimation)). To sum up, it was found that lack of health management, inappropriate body weight control, and health-related behaviors are related to body image misperception compared with real BMI. As such, it is necessary to have educational programs to encourage building proper perception of one’s body image and body weight, and to carry out health-related behaviors.

## 1. Introduction

Along with improved standard of living, obesity is on the rise, placing more burdens on the global community. According to the World Health Organization (WHO), the obese population increased by more than twofold between 1980 and 2014; approximately 13% (men 11%, women 15%) of the global adult population were obese, and 39% (men 39%, women 40%) were overweight as of 2016 [1]. Female adults’ prevalence of obesity has been on a steady rise, recording at 27.6% in 2009 and 28.4% in 2017 [2,3]. However, in a society where obesity is increasing, people tend to judge obesity based on weight rather than on scientific evidence, as the potential risks of obesity are highlighted, and the correct understanding of health and education program of obesity does not precede [4]. Social preference for thinner body image and the influence of mass media have made women perceive thinness as the ideal body image, leading them to misperceive their body to be overweight regardless of them actually being normal or underweight [5]. Body image perception refers to a subjective judgment of a person on his or her body regardless of medical or physiological indices [6]. Some epidemiological studies concerning obesity have used perceived body image as a proxy indicator of body weight status [7]. Thus, perceived body image can be divided into overestimation and underestimation. The influence of media has recently been leading to an increased overestimation of body image [8], with women showing higher perception of obesity and higher overestimation of body image than men [9], and also being less satisfied with their looks than men [10]. It is not only obese women, but also normal or underweight women who tend to overestimate themselves as being overweight or obese, inducing intensive stress, which brings about numerous health problems [11,12]. Furthermore, excessive concern toward weight and appearance may trigger problems such as anxiety, depression, and compulsive eating, thereby decreasing life quality.

The trend analysis on the body image perception of the KNHNES conducted from 2001 to 2013 showed that misperception was more seriously found among women and people at normal BMI [13]. Therefore, it is necessary to identify how adult women at normal BMI perceive their body image.

Obesity, a critical health problem caused by generic elements, plus lack of health management such as overeating and little exercise, is highly relevant to health-related behaviors [14]. Body image perception also plays an important role in managing obesity—when one perceives himself or herself as obese, health-related behaviors such as body weight management or exercises can be encouraged [15], and relevant elements as non-smoking and non-drinking [16] can be promoted as well. By analyzing health-related behaviors of women at normal BMI with overestimated or underestimated misperception of their body image, how their body images are perceived and how much of the health-related behavior they carry out can be identified.

According to the Statistics Korea (2010), 32.4% of Korean adults are obese [17], with women accounting for 28% of them. As the proportion of obese women at 25 kg/m^2^ or higher BMI has increased with age (twenties 12.1%, thirties 19.0%, forties 26.7%, fifties 30.8%, and sixties 43.3%), the prevalence of obesity has also soared [18]. Among the young age group (college students or economically active) with a high interest in their looks, women in their twenties and thirties had a higher rate of overestimating their body image as obese owing to the social atmosphere of one’s skills and values often being assessed through one’s looks, while those who are middle-aged and elderly actually showed higher obesity rates than their young counterparts [19]. In particular, while obesity is an important matter because it is directly related to health issues, overestimation of the body image may damage health owing to the subsequent excessive body weight management [20]. Therefore, it is necessary to distinguish between the body image perceptions and health-related behaviors of women aged 65 or above, and to compare each of them.

The study aims to utilize data from the KNHNES to check the correspondence between the actual body type and perceived body image of adult women aged 19 or above, and to identify the results by subdividing the differences in health-related behaviors (body weight management, drinking status, smoking status, exercising level, amount of sleep) based on their body image perceptions. Furthermore, by identifying the relationship between body image perception and health-related behaviors, the study aims to form proper values conducive to health management and the right recognition of one’s body type, and to be used as base data for the development of a health promotion program.

### Purposes of the Study

The first goal was to identify the characteristics of socio-demographic factors (age, educational level, marital status, employment status, household income) of women at normal BMI.

The second was to compare health-related behaviors (body weight management, exercising level, drinking status, smoking status, amount of sleep) of women at normal BMI with different body image perceptions.

## 2. Materials and Methods

### 2.1. Design

This study was conducted using data from the sixth Korea National Health and Nutrition Examination Surveys (KNHANES VI, 2013–2015), a nationwide cross-sectional survey conducted by the Korean Centers for Disease Control and Prevention (Seoul, Korea) to assess the health and nutritional status of the South Korean population. The 6th National Health and Nutrition Examination Survey was conducted with the approval of the Research Ethics Committee of the Korea Centers for Disease Control and Prevention (approval number 2013-07CON-03-4C for the 1st year; approval number 2013-12EXP-03-5C for the 2nd year), and the researcher downloaded the data on 22 January 2020 after obtaining approval to utilize the raw data of the National Health Nutrition Survey to the public. The Korea National Health and Nutrition Examination Survey (KNHNES) is a representative and reliable survey conducted nationwide on the health status, health awareness, and nutritional status of Koreans. The statistical result comprises a health examination, nutrition survey, and health interview, to be used as base data for the establishment and assessment of health policies such as objectives, evaluation, and development of health promotion programs of the Health Plan 2020. The KNHNES is a nationwide health and nutrition survey initiated under article 16 of the National Health Promotion Act established in 1995, which is a union of National Nutrition Survey and National Health and Health Behavior Survey that had been conducted separately. From 1998 to 2005, the KNHNES was conducted as a short-term tri-annual survey. After being converted to an annual survey in 2007, it has been conducted every year. Primary stratification categorizes provinces and cities, and secondary stratification categorizes general regions into 26 strata based on age and population ratio per age group, and apartment-clustered regions into 24 strata based on price per unit space and mean housing size per apartment complex; sample enumeration districts are calculated thereafter. Within the sample enumeration districts, 20 target households are sampled per district by adopting the systematic sampling method.

### 2.2. Participants

Among the 39,225 respondents of the Korea National Health and Nutrition Examination Survey (KNHNES) conducted from 2013 to 2017, 10,798 adult women with WHO BMI Classifications of overweight and obese for the Asia-Pacific at 18.5 or higher and below 25 were chosen, from whom 1120 pregnant and breast-feeding women and 391 women with no identified body image perception were excluded. Ultimately, a total of 9288 women were selected for the study.

### 2.3. Measures

#### 2.3.1. Body Mass Index (BMI)

BMI for the health examination of the KNHNES was calculated by dividing body weight (kg) by height (m^2^). Following the standard of the World Health Organization, the BMI groups were classified based on the BMI classifications for the Asian population: under 18.5 kg/m^2^ were classified as “underweight”, between 18.5 kg/m^2^ and 25 kg/m^2^ as “normal”, and 25 kg/m^2^ or above as “obese” [21].

#### 2.3.2. Subjective Body Image Perception

Subjects were asked, “What do you think of your body image?” and were instructed to answer in five scales: “severely thin”, “slightly thin”, “average”, “slightly overweight”, and “severely overweight”. This study re-categorized the five scales; “severely thin” and “slightly thin” were merged into “thin”, “average” was renamed as “normal”, and “slightly obese” and “severely obese” were merged into “obese”.

We classified the subjects of this study as follows. Among those with a normal BMI, a group that perceives its body type as obese: BIOP (body image over perception), a group that recognized it as normal: BICP (body image correct perception), and a group that recognized it as thin: BIUP (body image under perception).

#### 2.3.3. Health-Related Behavior (HRS)

Health-related behavior is a habit or an action intended to maintain and promote health. Among the seven health-related behaviors set by “Alameda 7” (7 to 8 h of sleep a day, having breakfast, no snacks, maintaining an appropriate weight, regular exercise, no alcohol, and no smoking [22]), six activities—smoking, drinking, body weight management, appropriate sleep (7 to 8 h), daily breakfast, and muscular or aerobic exercise—were used for the study; subjects with 0 to 2 points were classified as the non-health practice group, and those with 3 to 6 points were classified as the health practice group.

### 2.4. Statistical Analyses

For statistical analyses, we adopted SAS ver. 9.4 (SAS Institute Inc, Cary, NC, USA), and conducted a complex samples design following the KNHNES’s guideline on raw data. First, frequency and percentage (%) were used to identify the body image perception of Korean women at normal BMI. Second, cross tabulation was conducted to compare body image perceptions among different general characteristics of Korean women at normal BMI. Third, cross tabulation and generalized linear model (GLM) were carried out to compare health-related behaviors of women (normal BMI) with different body image perceptions. Furthermore, the levels of health-related behaviors were defined as high and low, and the odds ratio regarding high-level health-related behaviors of women (normal BMI) with different body image perceptions was calculated.

## 3. Results

### 3.1. Body Shape According to General Characteristics

It was found that 87.6% (*n* = 8039, weighted *n* = 4,261,224), approximately most of adult Korean women, misperceived their body image. Specifically, 33.0% (*n* = 1249, weighted *n* = 603,721) of women at normal BMI overestimated themselves as obese, and 12.4% (*n* = 2911, weighted *n* = 1,603,279) underestimated themselves as thin (Figure 1).

The percentage of women with overestimated body image perception differed by age group: those in their twenties, thirties, and forties were at 37%; those in their fifties were at 31.6%; while those in their sixties and seventies or above were at 28.3% and 11.6%, respectively. In short, young women had a higher tendency of overestimating their body image perception as obese when compared with their actual body weight. In regards to the marital status, unmarried women recorded 37.1% of overestimated perception, which is higher than the other group. Regarding household income, a higher income quintile meant higher overestimated perception. For type of occupation, those working in the service and sales sector had 39.6% of overestimated body image perception, and office workers followed with 37.6%—both of which are relatively higher than other types. Moreover, those in the agricultural and fishery sector showed a lower rate than others, at 24.1%. In terms of educational status, high-school graduates showed 38.2% and college graduates were at 34%; a higher educational status led to higher overestimated body image perception. Menopausal women had higher overestimated body image perception of 36.6%, higher than 26.1% of their counterparts (Table 1).

### 3.2. Health-Related Behaviors of Different Body Image Perceptions

Subjects aged 64 or under showed statistically significant difference in body weight management, smoking status, exercising level, and amount of sleep depending on the different body image perception.

About 28% of the BIUP (Body Image Underestimation Perception) group responded yes to the question asking any attempted body weight management in the recent one year, while only 13.5% of the BIOP (Body Image Overestimation Perception) group said yes. Also, 67.6% of the BIOP group said they tried to reduce weight, which the figure was higher than other groups; 16.5% of the BIOP group responded they tried to gain weight, which was also found to be higher than other groups.

The BIUP group had less aerobic exercises only which 46.1% said they do; the percentage was 54% for the BIOP group and 51.5% for the BICP (Body Image Correct Estimation Perception) group. Meanwhile, in terms of muscular exercise, 13% of the BICP group said yes, followed by 16.5% of the BIUP group, and 15.4% of the BIOP group. For smoking, 88.8% of the BICP group responded they never smoke, followed by 87.6% of the BIUP group and 85.9% of the BIOP group. The BIOP group was the top with 41.7% in terms of sleeping below 7 h. Also, 28.9% said they sleep between 7 to 8 h—the recommended sleeping hours—which was less compared to the other groups (Table 2).

Subjects aged 65 or above showed statistically significant difference depending on the body image perception in terms of attempts to manage body weight within the recent year, exercises (aerobic), and smoking status. Regarding the body weight management question, 16.3% of the BICP group said yes, which was higher than other groups. Also, 20.4% of the BIUP group said they are attempting to gain weight, which was also higher than other groups. On the other hand, 44.7% of the BIOP group said they are trying to lose weight, which was higher than others. In short, when one misperceives his or her body image regardless of age, those underestimated tried to gain weight while those overestimated worked to control weight, both of which showed higher tendencies compared to the others. In terms of aerobic exercising, 22.2% of the BIUP group said they do, which was less than the other groups. Regarding smoking, 93.9% of the BICP said they do not smoke; the percentage was 90.2% for the BIUP group and 93.7% for the BIOP group (Table 3). In addition, it was found that the body-image-misperceived groups had higher rate of smoking compared to the correctly perceived group. Meanwhile, among those aged 64 or below, the BICP group had higher rate of doing aerobic and muscular exercises; the BICP group aged 65 or more had less tendency of exercising. Also, the BIOP group aged 64 or below showed less sleeping time compared to their counterparts.

### 3.3. Odds Ratio in Regards to Health-Related Behaviors

It was found that 55.8% (*n* = 4201, weighted *n* = 2,330,958) of those aged 64 or below were engaging in health-related behaviors; 39.4% (*n* = 752, weighted *n* = 271,120) of those aged 65 or above were identified to be putting effort into those activities. With regard to health-related behaviors among subjects aged 64 or under, the BIUP group compared with the BICP group showed a statistically significant odds ratio of 0.718 (confidence interval (C.I.) 0.594–0.866), while the BIOP group showed the ratio of 0.961 (C.I. 0.853–1.083), which is not statistically significant. Among the subjects aged 65 or above, the BIUP group compared with the BICP group showed an odds ratio of 0.572 (C.I. 0.443–0.738), which is statistically significant, while the BIOP group showed an odds ratio of 1.683 (C.I. 1.260–2.248), which is also statistically significant. In other words, the body-image-underestimated group was less engaged in health-related behaviors, but if those above the age of 65 perceived themselves as obese, they tended to carry out more of those activities (Table 4).

This section may be divided by subheadings. It should provide a concise and precise description of the experimental results, their interpretation, as well as the experimental conclusions that can be drawn.

## 4. Discussion

The study aimed to identify the differences between subjective body image perceptions of Korean women at normal BMI with regard to health-related behaviors.

About 87.6% of the subjects aged between their twenties and seventies were found to misperceive their body image as thin or obese even though they were at normal BMI. Moreover, the misperception rate was higher at a younger age. In addition, in terms of the types of occupation, those working in the service and sales sector showed the highest overestimation of their body image. This was driven by the fact that women tend to have different ideas between their actual body type and the perceived body image [23]. In particular, the result was identical to the research [24] in that younger age led to higher body image overestimation. The reason for this is considered to be the social atmosphere prevalent across the world that regards thinness as the major yardstick for beauty; as women in the service sector tend to be more sensitive to how they are shown to others, it is deemed that their overestimation is higher than other types of occupation [25].

If women at normal BMI build fear of obesity and overestimate their body image owing to the biased social atmosphere that values thinness, it can lead to an excessive control of body weight and irregular and imbalanced diet, not only threatening individual health, but ultimately triggering psychological and emotional issues—all of which are emerging as social issues [25,26,27]. Unlike how all age groups were reported to be overestimating their body image in previous research [23,24,25,26,27,28,29,30,31], those in their seventies and above were found to have higher underestimation than overestimation in this study. To reflect age-specific singularity on the body image perception, this study analyzed body image perceptions by dividing the subjects into two age groups: under 65 and 65 or above.

This study found that about 50% of women aged 64 or below attempted to lose weight during the past year, based on the criteria for health-related behaviors. In terms of smoking, the misperceived groups had a higher percentage for “I sometimes smoke or I smoke everyday”; in particular, the overestimated group showed the highest percentage of smoking sometimes or everyday compared with the other groups. Among the subjects aged 65 or above, the BIUP group showed the highest smoking rate. The results are relevant to preceding research [32] stating that the high smoking rate of the misperceived groups is driven by stress, anxiety, unsatisfying body image, low self-esteem [33,34] in spite of their efforts to have ideal body images [35,36], as well as worries over weight gain. Moreover, it was found that misperceptions adversely impact health-related behaviors.

Exercising is an important physical activity that enhances body image by helping one to achieve his or her self-realization of becoming slim, while providing health benefits [37]. As the study analyzed how much aerobic exercise the subjects were carrying out, the BIOP group was found to be engaged in more aerobic exercises compared with the BICP group across all the age groups; the BIUP group had the lowest percentage. The BIUP group, in particular, had a lower figure than the average percentage of Korean women doing aerobic exercises (47.6% for those aged 64 or under, and 26.9% for those aged 65 or above) [38]. Additionally, as preceding research studied how often Korean adults engaged in physical activities in a week based on their BMI, it found that middle-level physical activities were carried out more often by underweight subjects compared with those at normal BMI; overweight and obese subjects were found to be doing more exercises. Furthermore, the number of walking days was also high in the order of underweight, normal, overweight, and obese groups, similar to the result of this study [39]. Such a result implies that one’s body image perception has a direct impact on health-related behaviors such as exercise. In short, if women with normal BMI overestimated their body image, they conducted aerobic exercises for health; while if they underestimated their body image, the frequency of exercising fell short to the average rate of Koreans doing aerobic exercises. It is highly important for seniors to build muscular strength to prevent falling and enhance mobility [40]. The BICP group aged 64 or under was at 18.3%, followed by 16.5% of the BIUP, and 15.4% of the BIOP group in terms of doing muscular exercises; among those aged 65 or above, the BIOP group showed 10.4%, followed by 9.5% of the BICP, and 6.7% of the BIUP group. The BIUP group, in particular, was found to be carrying out less muscular exercise than the average 9.0% of Korean women aged 65 or above [38]. Lack of muscular exercise reduces muscle mass, which in turn leads to depression, insulin resistance, and cardiovascular diseases [41,42,43], having significant repercussions for the elderly. Therefore, it is safe to say that, as the BIOP group aged 64 or below and the BIUP group aged 65 or more are conducting less muscle exercises among the subjects with misperceived body image, they have a higher change of reduced muscle mass and decline in muscular power as compared with the BICP group.

Meanwhile, the BIOP group aged 64 or above had the highest rate of aerobic exercises, while those aged 65 or above were found to be conducting muscular exercises most often. A research that identified the relationship among muscle mass, muscular strength, and exercise (frequency, hours) with 274 subjects aged 65 or above came to a conclusion that the three elements had a positive correlation [44]. It is, therefore, considered that the muscle mass and muscular strength of the BIOP group have affected the subjects’ engagement in an exercise. However, the causes behind this need to be identified in future research. Physical activities prevent obesity and chronic diseases, and enhance physical image, self-esteem, and quality of life. If one maintains a healthy body type through physical activities, it will not only improve one’s health, but will also be beneficial for the nation by reducing government spending on public health [45,46]. In terms of sleeping hours, the BIOP group showed a higher figure for the “less than 7 h of sleep” criterion compared with the BICP group across all the age groups. Previous research on sleeping hours showed a negative correlation, with fewer hours of sleep leading to a higher risk of obesity [47,48,49,50]. Another study reported that a group with 7 h or less hours of sleep showed increased BMI and greater waist circumference, body weight, and body fat [49,51]; this study also showed that the BIOP group was sleeping less than the other groups. This means that the lack of sleep is not merely a matter of number, but a physical issue that can lead to increased BMI and waist circumference, just as shown in preceding research. Therefore, it may work as a trigger wherein the misperceived body image can eventually become the real body type, meaning that it could be a risk factor of obesity for people with normal weight. The reason for these results is suspected that persisted irregular sleeping patterns and lack of sleep make a person unsatisfied with his or her health-related lifestyle, which in turn affects the body image, and ultimately leads him or her to overestimate his or her body image, despite being at normal BMI.

Among the seven health-related behaviors set by “Alameda 7”, six activities—smoking, drinking, body weight management, appropriate sleep, daily breakfast, and muscular or aerobic exercise (omitting having snacks, as it is not the survey criteria of the KNHNES)—were used to divide the subjects into two groups: the non-health practice group and health practice group. Then, the odds ratio analysis was conducted on these groups. As a result, all the groups with misperceived body image aged 64 or under were not conducting health-related behaviors compared with the BICP group; the BIUP aged 65 or above were not carrying out those activities compared with the BICP group, and the BIOP group had the highest engagement in the activities. In short, among the body-image-misperceived groups, those aged 64 or under had less engagement in health-related behaviors, but the BIOP group of the other age range had high engagement in the activities. Such a result might be because of the fact that the social climate of preferring thinness may have also affected the seniors, as well as the fact that obesity is directly linked to physical conditions such as cardiovascular diseases and joint problems for those aged 65 or more, leading them to do more physical activities.

The limitations of this study are that only people with a normal BMI were studied. Further research will require research design for those with an abnormal BMI, as body type recognition of people with abnormal BMI is also an important area for health problems.

To sum up, we identified that lack of health management, inappropriate body weight control, and health-related behaviors are related to body image misperception compared with the real BMI. Such a result emphasizes the importance of subjective body image perception with regard to women’s health, and should be utilized as significant scientific evidence for health management of women with normal BMI, and stresses the necessity of correctly perceiving body image and changing daily habits.

## 5. Conclusions

This study was conducted to seek measures for correct body image perceptions and proper health management by comparing health-related behaviors of different subjective body image perceptions held by adult women at normal BMI (based on WHO’s BMI classifications of overweight and obesity for the Asia-Pacific) among the participants of the 6th and 7th KNHNES (2013–2017). As a result, 33.0% of women at normal BMI overestimated themselves as obese, and 12.4% underestimated themselves as thin; 87.6%, approximately most of adult Korean women, misperceived their body image. Moreover, as it was found that, across all the age groups, those who were misperceiving their body image were more engaged in wrong health-related behaviors such as non-physical activities, smoking, insufficient sleep, and excessive body weight control, it can be seen that body image misperception is linked to health-related behaviors.

To sum up, among the subjects, those who misperceived their body image were found to be less engaged in health-related behaviors compared with those with correct perception, while having higher risks of undermining health. As such, it is necessary to have educational programs to encourage building proper perception of one’s body image and body weight, and carry out health-related behaviors. Moreover, it is considered that the establishment of a social climate and measures to help people build healthy habits to maintain and enhance health are needed.

## Figures and Tables

**Figure 1 ijerph-17-05534-f001:**
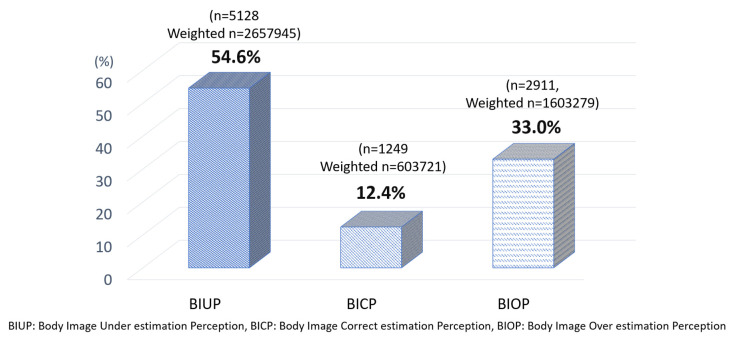
Body image perceptions of normal body mass index (BMI) females.

**Table 1 ijerph-17-05534-t001:** Body image perceptions—differences according to general characteristics.

Characteristic	Variable Item	BICP	Misperceiving Body Image		Total	Rao–Scott χ^2^(*p*)
			BIUP	BIOP		
Age (years)	20–29	680 (53.9)	95 (8.2)	471 (37.9)	1246 (100)	428.992 (<0.001)
	30–39	958 (54.4)	132 (8.5)	635 (37.2)	1725 (100)	
	40–49	1089 (53.4)	176 (9)	765 (37.6)	2030 (100)	
	50–59	970 (56)	208 (12.3)	539 (31.6)	1717 (100)	
	60–69	736 (56.1)	218 (15.5)	346 (28.3)	1300 (100)	
	≥70	695 (55.4)	420 (33)	155 (11.6)	1270 (100)	
	Total	5128 (54.6)	1249 (12.4)	2911 (33)	9288 (100)	
Marital status	Never married	795 (54.4)	117 (8.6)	525 (37.1)	1437 (100)	145.374 (<0.001)
	Married (cohabit)	3517 (55.2)	745 (11.3)	2045 (33.6)	6307 (100)	
	Married (separated)	812 (52.4)	386 (23.9)	339 (23.7)	1537 (100)	
	Total	5124 (54.6)	1248 (12.4)	2909 (33)	9281 (100)	
Household income	Q1	854 (53.4)	405 (23.5)	323 (23.1)	1582 (100)	156.437 (<0.001)
	Q2	1181 (53.3)	289 (12.6)	714 (34.1)	2184 (100)	
	Q3	1427 (54.1)	285 (10.6)	890 (35.3)	2602 (100)	
	Q4	1642 (56.7)	262 (9)	972 (34.4)	2876 (100)	
	Total	5104 (54.6)	1241 (12.4)	2899 (33)	9244 (100)	
Occupation	Professional, manage	738 (58.6)	109 (9.7)	397 (31.8)	1244 (100)	82.784 (<0.001)
	Office worker	477 (54.5)	68 (8)	331 (37.6)	876 (100)	
	Service and sales	685 (51.5)	121 (9)	491 (39.6)	1297 (100)	
	Agricultural	148 (50.7)	75 (25.1)	56 (24.1)	279 (100)	
	Technician, manipulation	138 (55)	25 (12.6)	80 (32.4)	243 (100)	
	Simple labor	401 (51.5)	143 (15.8)	218 (32.6)	762 (100)	
	Inoccupation	2343 (55.8)	622 (13.5)	1233 (30.7)	4198 (100)	
	Total	4930 (54.9)	1163 (12)	2806 (33)	8899 (100)	
Education	Elementary school	1001 (54.9)	493 (26.2)	324 (18.8)	1818 (100)	299.196 (<0.001)
	Middle school	443 (54.5)	117 (14)	247 (31.5)	807 (100)	
	High school	1600 (52.6)	270 (9.2)	1095 (38.2)	2965 (100)	
	≥College	1884 (57.1)	283 (8.9)	11,309 (34)	3306 (100)	
	Total	4928 (54.9)	1163 (12.1)	2805 (33)	8896 (100)	
Menopause	No	2973 (54.2)	485 (9.3)	1968 (36.6)	5426 (100)	157.41 (<0.001)
	Yes	2155 (55.5)	764 (18.3)	943 (26.1)	3862 (100)	
	Total	5128 (54.6)	1249 (12.4)	2911 (33)	9288 (100)	

BIOP—body image over estimation perception; BICP—body image correct estimation perception; BIUP—body image under estimation perception.

**Table 2 ijerph-17-05534-t002:** Health-related behaviors of different body image perceptions (aged ≤ 65).

Characteristic	Variable Item	BICP	Misperceiving Body Image		Total	Rao–Scott χ^2^(*p*)
			BIUP	BIOP		
Weight control attempt during the past 1 year	Lose weight	1748 (44.7)	100 (14.8)	1739 (67.6)	3587 (50)	1030.68 (<0.001)
	Maintain weight	1179 (28.4)	198 (28.2)	360 (13.5)	1737 (23)	
	Gain weight	52 (1.2)	121 (16.5)	4 (0.1)	177 (2.3)	
	Nothing to control weight	1112 (25.7)	286 (40.5)	518 (18.8)	1916 (24.6)	
	Total	4091 (100)	705 (100)	2621 (100)	7417 (100)	
Aerobic exercise	No	1662 (48.5)	325 (53.9)	1030 (46)	3017 (48.1)	9.069 (0.011)
	Yes	1669 (51.5)	263 (46.1)	1125 (54)	3057 (51.9)	
	Total	3331 (100)	588 (100)	2155 (100)	6074 (100)	
Muscle strengthening exercise	No	3242 (81.7)	566 (83.5)	2161 (84.6)	5969 (82.9)	6.812 (0.033)
	Yes	703 (18.3)	107 (16.5)	362 (15.4)	1172 (17.1)	
	Total	3945 (100)	673 (100)	2523 (100)	7141 (100)	
Smoking status	Current smoker	188 (5.0)	49 (7.5)	178 (7.5)	276 (4.1))	17.391 (0.008)
	Former smoker	230 (6.3)	38 (5.1)	163 (6.6)	431 (6.3)	
	Never smoker	3673 (88.8)	620 (87.6)	2280 (85.9)	6573 (87.6)	
	Total	4091 (100)	706 (100)	2620 (100)	7417 (100)	
Smoking	Total	7.40 ± 0.45	8.25 ± 1.07	8.44 ± 0.47	7.96 ± 0.33	1.45 (0.236)
Alcohol consumption	None	623 (15.3)	135 (19.1)	393 (15.2)	1151 (15.6)	15.497 (0.115)
	>1 times/month	1000 (26.8)	182 (27.6)	687 (28.6)	1869 (27.5)	
	1 times/month	536 (15.1)	85 (14.1)	313 (12.8)	934 (14.2)	
	2–4 times/month	976 (27.9)	145 (25.6)	659 (27.7)	1780 (27.6)	
	2–3 times/week	453 (12.5)	65 (11)	278 (12.2)	796 (12.2)	
	≥4 times/week	88 (2.5)	17 (2.6)	86 (3.5)	191 (2.9)	
	Total	3676 (100)	629 (100)	2416 (100)	6721 (100)	
Drinking	1–2 glasses/time	1623 (50.9)	284 (53.3)	936 (43.3)	2843 (48.3)	53.517 (<0.001)
	3–4 glasses/time	758 (24.8)	115 (25.3)	486 (23.3)	1359 (24.3)	
	5–6 glasses/time	379 (13)	50 (11.1)	282 (14.9)	711 (13.5)	
	7–9 glasses/time	169 (6.6)	30 (6.8)	166 (9.3)	365 (7.6)	
	≥10 glasses/time	124 (4.7)	15 (3.5)	153 (9.2)	292 (6.3)	
	Total	3053 (100)	494 (100)	2023 (100)	5570 (100)	
Sleep	>7 h/day	1515 (37.2)	269 (36.1)	1071 (41.7)	2855 (38.7)	14.203 (0.007)
	>8 h/day	1271 (32.2)	230 (35)	762 (28.9)	2263 (31.2)	
	≤8 h/day	1246 (30.6)	190 (28.9)	745 (29.4)	2181 (30)	
	Total	4032 (100)	689 (100)	2578 (100)	7299 (100)	

**Table 3 ijerph-17-05534-t003:** Health-related behaviors of different body image perceptions (aged >65).

Characteristic	Variable Item	BICP	Misperceiving Body Image		Total	Rao–Scott χ^2^(*p*)
			BIUP	BIOP		
Weight control attempt during the past 1 year	Lose weight	132 (12.9)	14 (2.1)	132 (44.7)	278 (14.7)	294.809 (<0.001)
	Maintain weight	176 (16.3)	47 (9.4)	39 (13.8)	262 (13.9)	
	Gain weight	50 (5.1)	117 (20.4)	9 (2.9)	176 (9.2)	
	Nothing to control weight	677 (65.7)	360 (68)	110 (38.6)	1147 (62.2)	
	Total	1035 (100)	538 (100)	290 (100)	1863 (100)	
Aerobic exercise	No	668 (67.5)	370 (77.8)	191 (67.4)	1229 (70.3)	12.707 (0.002)
	Yes	303 (32.5)	113 (22.2)	90 (32.6)	506 (29.7)	
	Total	971 (100)	483 (100)	281 (100)	1735 (100)	
Muscle strengthening exercise	No	891 (90.5)	458 (93.3)	261 (89.6)	1610 (91.1)	2.945 (0.229)
	Yes	95 (9.5)	34 (6.7)	23 (10.4)	152 (8.9)	
	Total	986 (100)	492 (100)	284 (100)	1762 (100)	
Smoking status	Current smoker	27 (2.8)	30 (6.3)	6 (1.3)	63 (3.6)	16.701 (0.002)
	Former smoker	33 (3.4)	23 (3.5)	13 (5)	69 (3.7)	
	Never smoker	977 (93.9)	489 (90.2)	271 (93.7)	1737 (92.8)	
	Total	1037 (100)	542 (100)	290 (100)	1869 (100)	
Smoking		11.18±1.79	10.09±2.04	11.47±3.14	10.68±1.29	0.110 (0.900)
Alcohol consumption	None	229 (36.9)	136 (44.9)	75 (33.9)	440 (38.5)	10.028 (0.438)
	>1 times/month	206 (31.6)	84 (27.9)	59 (32.2)	349 (30.7)	
	1 times/month	77 (12.7)	30 (9.8)	25 (13.6)	132 (12.1)	
	2–4 times/month	75 (11)	28 (8.3)	23 (12.9)	126 (10.6)	
	2–3 times/week	31 (4.8)	9 (4.8)	11 (5.9)	51 (5)	
	≥4 times/week	20 (3)	13 (4.4)	4 (1.6)	37 (3.1)	
	Total	638 (100)	300 (100)	197 (100)	1135 (100)	
Drinking	1–2 glasses/time	339 (83.1)	136 (84.7)	101 (82.6)	576 (83.4)	0.258 (0.992)
	3–4 glasses/time	48 (12.6)	21 (11.6)	17 (12.8)	86 (12.4)	
	≥5 glasses/time	21 (4.2)	7 (3.7)	4 (4.7)	32 (4.2)	
	Total	408 (100)	164 (100)	122 (100)	694 (100)	
Sleep	>7 h/day	450 (42.9)	236 (43.2)	121 (43.2)	807 (43.1)	2.725 (0.605)
	>8 h/day	244 (24.2)	93 (20.5)	71 (25)	408 (23.3)	
	≤8 h/day	311 (32.8)	185 (36.3)	92 (31.8)	588 (33.7)	
	Total	1005 (100)	514 (100)	284 (100)	1803 (100)	

BIOP—body image over estimation perception; BICP—body image correct estimation perception, BIUP—body image under estimation perception.

**Table 4 ijerph-17-05534-t004:** Odds ratio regarding high-level health-related behaviors.

Group	BICP	Misperceiving Body Image	
		BIUP	BIOP
≤64 aged	1	0.718(0.594–0.866)	0.961(0.853–1.083)
>65 aged	1	0.572(0.443–0.738)	1.683(1.260–2.248)

BIOP—body image over estimation perception; BICP—body image correct estimation perception, BIUP—body image under estimation perception.

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
