# Peer review of "Analysis of Health-Related Behaviors of Adult Korean Women at Normal BMI with Different Body Image Perceptions: Results from the 2013–2017 Korea National Health and Nutrition Examination Survey (KNHNES)"

_ijerph, 2020, doi:10.3390/ijerph17155534_

Round 1

Reviewer 1 Report

The authors have conducted a very interesting study.

I put my feedback into specific line-by-line comments below.

In section 2.3.2. Subjective body image perception please explain on what grounds the selected subjects were added to the BIUP, BICP and BIOP groups. Please add the average BMI for these groups

Results

Please increase the font size of the key in Figure 1.

According to Fig. 1, 87,6% of respondents incorrectly assess their appearance, while first sentence- lines 164-165- contains data that does not support this claim, namely: “ It was found that 45.4% (n=5,128, weighted n=2,657,945), approximately half of 165 adult Korean women were misperceiving their body image.” The same appears in discussion in line 242. Please explain it.

I guess that in Tab1 the data is presented in units - quantitative (numeric) and%. Please add units - because in the present form the results are ambiguous.

Please correct the abbreviation of “Body Image Correct Estimation Perception” (BCIP), either use  BICEP or remove the word "Estimation" from the phrase.

At the end of the sentence in line 251 please add reference. You can add- Michalczyk, M.M.; Zajac-Gawlak, I.; Zając, A.; Pelclová, J.; Roczniok, R.; Langfort, J. Influence of Nutritional Education on the Diet and Nutritional Behaviors of Elderly Women at the University of the Third Age. Int J Environ Res Public Health. 2020, Jan 21;17(3). pii: E696. doi: 10.3390/ijerph17030696.

Conclusion:

The conclusion introduces new information that has not been previously presented in the manuscript.  It must be edited

Author Response

▶In section 2.3.2. Subjective body image perception please explain on what grounds the selected subjects were added to the BIUP, BICP and BIOP groups. Please add the average BMI for these groups

▶ For the readers’ understanding, we inserted the following sentence in Section 2.3.2.

<page 4 line 142-145>We classified the subjects of this study as follows. Among those with a normal BMI, a group that perception its body type as obese: BIOP (Body Image Over estimation Perception), a group that recognized it as normal: BICP (Body Image Correct estimation Perception), and a group that recognized it as thin: BIUP (Body Image Under estimation Perception).

▶Results: Please increase the font size of the key in Figure 1.

▶Yes, I fixed it.

▶According to Fig. 1, 87,6% of respondents incorrectly assess their appearance, while first sentence- lines 164-165- contains data that does not support this claim, namely: “ It was found that 45.4% (n=5,128, weighted n=2,657,945), approximately half of 165 adult Korean women were misperceiving their body image.” The same appears in discussion in line 242. Please explain it.

▶Thank you for your careful review.
There was a mistake in our calculation of percentage. The text was corrected by correct calculation.

<page 4 line 169-170>
It was found that 87.6% (n=8,039, weighted n=4,261,224), approximately most of adult Korean women were misperceiving their body image.

▶I guess that in Tab1 the data is presented in units - quantitative (numeric) and%. Please add units - because in the present form the results are ambiguous.

▶For accurate interpretation, the units (n %) are inserted in Table 1.

▶Please correct the abbreviation of “Body Image Correct Estimation Perception” (BCIP), either use BICEP or remove the word "Estimation" from the phrase.

▶I agree.
The abbreviation (BIOP, BICP, BIOP) removed the word for the estimate because the body has a sufficient description of the estimate.

▶At the end of the sentence in line 251 please add reference. You can add- Michalczyk, M.M.; Zajac-Gawlak, I.; Zając, A.; Pelclová, J.; Roczniok, R.; Langfort, J. Influence of Nutritional Education on the Diet and Nutritional Behaviors of Elderly Women at the University of the Third Age. Int J Environ Res Public Health. 2020, Jan 21;17(3). pii: E696. doi: 10.3390/ijerph17030696.

▶ Ok. refer. insert: No. 25 insertion

▶Conclusion: The conclusion introduces new information that has not been previously presented in the manuscript.  It must be edited

▶Thank you for your careful and high-quality opinion, and for helping our paper improve.

Reviewer 2 Report

This is a very interesting topic. However, there are a few points that need to be addressed:

  1. The manuscript needs extensive English-language copy editing.
  2. Lines 47-48, the authors write, “However, despite the negative social atmosphere towards obesity, only its potential risks are being highlighted.” Are there benefits? This sentence should be re-written or removed.
  3. In line 74 the authors write, “As the proportion of obese women at 25kg/m² or higher…” This is not the definition of obesity. Obesity is BMI≥30 kg/m2. BMI 25kg/m² to < 30 kg/m2 is overweight. Please correct this terminology.
  4. The Alameda County Study did not identify no alcohol as a habit of healthy people; rather, it was drinking less than 5 drinks in one sitting.
  5. The comparison of body image by menopause is confounded by the age of the respondents. It would be interesting to repeat that comparison among women aged 45-60 by menopause status (post-menopause yes/no).
  6. The paragraph on health-related behaviors of different body image perceptions is very confusing and counter-intuitive. This information is presented in a table and really does not need to be re-iterated in the text.

Author Response

  1. The manuscript needs extensive English-language copy editing.

    I commissioned a company specializing in editing.
  2. Lines 47-48, the authors write, “However, despite the negative social atmosphere towards obesity, only its potential risks are being highlighted.” Are there benefits? This sentence should be re-written or removed.

    The intention of this sentence is to recognize oneself as obese even though one's weight is normal, because only the risk of obesity is highlighted and there is no education on health care or the definition of obesity.
    To help understand this sentence, the sentence was modified in a kind and detailed manner as follows.

    <page 2 line 45-48>

    However, in a society where obesity is increasing, people tend to judge obesity based on weight rather than negative perception of obesity based on scientific evidence, as the potential risks of obesity are highlighted, and the correct understanding of health and education program of obesity does not precede.
  3. In line 74 the authors write, “As the proportion of obese women at 25kg/m² or higher…” This is not the definition of obesity. Obesity is BMI≥30 kg/m2. BMI 25kg/m² to < 30 kg/m2 is overweight. Please correct this terminology.

    We used the BMI criteria for Asian standards.
    In line 129  (World Health Organization. The Asia-Pacific perspective: redefining obesity and its treatment. Sydney: Health Communications Australia; 2000. Regional Office for the Western Pacific)

  4. The Alameda County Study did not identify no alcohol as a habit of healthy people; rather, it was drinking less than 5 drinks in one sitting.

    Alameda's drinking standard is less than five cups, but this study applied stricter drinking criteria to health activities as secondary data using national data could not be selected for those who drank less than five cups.
  5. The comparison of body image by menopause is confounded by the age of the respondents. It would be interesting to repeat that comparison among women aged 45-60 by menopause status (post-menopause yes/no).

    We agree with you. However, our subjects were studied and designed throughout women’s lives, and menopause could occur regardless of age due to various health problems (e.g., uterine cancer, breast cancer, or hormonal problems), and in this case, we did not limit age because it could be particularly sensitive about body type. However, further studies agree to limit the age limit for menopause.We agree with you.

  6. The paragraph on health-related behaviors of different body image perceptions is very confusing and counter-intuitive. This information is presented in a table and really does not need to be re-iterated in the text.

    Section 3.2 has been deleted to understand the contents of the journal clearly.

Reviewer 3 Report

The authors aimed to provide basic data to seek solutions for ideal physical activities and right body image perception by comparing health-related behaviors of women at normal BMI and their blood lipid profile. The study is very important for the proper perception of our body image, whereas minor revision should be considered;

  • The readability and syntax of the manuscript will be substantially improved if it is reviewed by a formal translation agency or by a colleague whose first language is English.
  • The abstract should be more concise without sacrificing important differential results of health-related behaviors and different BMI in different ages. (Results should be more detailed).
  • The introduction lacked to the hypothesis.
  • In subjective body image perception, How did the authors validate and check the reliability of this measure? 
  • In statistical analysis, How did the authors assess the normality of the data distribution? Also, How did they determine the significance of their results?
  • Did the authors estimate the sample size?
  • The main limitation of the study design should be addressed.
  • Many old references should be updated (for example, No 10, 16, 21, 40). 

Author Response

  1. The readability and syntax of the manuscript will be substantially improved if it is reviewed by a formal translation agency or by a colleague whose first language is English.|
    I commissioned a company specializing in editing.
  2. The abstract should be more concise without sacrificing important differential results of health-related behaviors and different BMI in different ages. (Results should be more detailed).

    We corrected.

    <Line 25-35> “Results showed that 87.6% approximately most of adult Korean women misperceived their body image. Misperception of body image was related to inappropriate health-related behaviors such as smoking, insufficient sleeping, and excessive body weight management; those who had underestimated their body image (≤64 OR (0.718(C.I. 0.594-0.866))) carried out fewer health-related behaviors, while women aged 65 or above engaged in more health-related behaviors when they perceived themselves as obese (OR 1.683(C.I. 1.260-2.248; overestimation)). To sum up, it was found that lack of health management, inappropriate body weight control, and health-related behaviors are related to body image misperception compared to the real BMI. As such, it is necessary to have educational programs to encourage building proper perception of one’s body image and body weight, and to carry out health-related behaviors.”

  3. The introduction lacked to the hypothesis.

    Complements the hypothesis.

    <Line 53> “some epidemiological studies concerning obesity have used perceived body image as a proxy indicator of body weight status”

    <Line 59> “Furthermore, excessive concern toward weight and appearance may trigger problems such as anxiety, depression, and compulsive eating, thereby decreasing life quality.

  4. In subjective body image perception, How did the authors validate and check the reliability of this measure?
    It is a self-written questionnaire that asks for individuals’ subjective views.
    Quality is ensured by data that the Korea Centers for Disease Control and Prevention research provides.
  5. In statistical analysis, How did the authors assess the normality of the data distribution? Also, How did they determine the significance of their results?
    No analysis has been conducted that requires the assumption of normality.
    Therefore, a separate normality test was not covered.

  6. Did the authors estimate the sample size?
    Since this study used data provided by research and provided by the Korea Centers for Disease Control and Prevention, a separate sample size calculation was not performed.

  7. The main limitation of the study design should be addressed.
    The following is added to Page 9 line 303-305.
    This study only studied the body type of normal BMI people. However, further research will require research design for abnormal BMI people, as body type recognition of people with abnormal BMI is also an important area for health problems.

  8. Many old references should be updated (for example, No 10, 16, 21, 40).

    10. <Delete>
    16. <Change> Lopez Khoury, E. N., Litvin, E. B., Brandon, T. H., 2009. The effect of body image threat on smoking motivation among college women: Mediation by negative affect. Psychology of Addictive Behaviors, 23(2), 279.
    21. “BMI Asia Pacific Standards” – This reference is original.
    40. <Change> Benichou, O., Lord, S. R., 2016. Rationale for strengthening muscle to prevent falls and fractures: a review of the evidence. Calcified tissue international, 98(6), 531-545.
